# Predictors of time to recovery and non-response during outpatient treatment of severe acute malnutrition

**Suvi T. Kangas**[1,2]*, **Cécile Salpéteur**[2], **Victor Nikièma**[3], **Christian Ritz**[1], **Henrik Friis**[1], **André Briend**[1,4], **Pernille Kaestel**[1]

**1** Department of Nutrition, Exercise and Sports, University of Copenhagen, Copenhagen, Denmark, **2** Expertise and Advocacy Department, Action Against Hunger (ACF), Paris, France, **3** Nutrition and Health Department, Action Against Hunger (ACF) Mission in Burkina Faso, Paris, France, **4** Center for Child Health Research, University of Tampere School of Medicine, Tampere University, Tampere, Finland

* suvi_kangas@hotmail.com

**Data Availability Statement:** The full dataset used to produce the current manuscript is available from the ZENODO platform: https://zenodo.org/record/

## Abstract

### Background

Every year, over 4 million children are treated for severe acute malnutrition with varying program performance. This study sought to explore the predictors of time to recovery from and non-response to outpatient treatment of SAM.

### Methods

Children with weight-for-height z-score (WHZ) <-3 and/or mid-upper arm circumference (MUAC) <115 mm, without medical complications were enrolled in a trial (called MANGO) from outpatient clinics in Burkina Faso. Treatment included a weekly ration of ready-to-use therapeutic foods. Recovery was declared with WHZ ≥-2 and/or MUAC ≥125 mm, for two weeks without illness. Children not recovered by 16 weeks were considered as non-response to treatment. Predictors studied included admission characteristics, morbidity and compliance during treatment and household characteristics. Cox proportional hazard models were fitted and restricted mean time to recovery calculated. Logistic regression was used to analyse non-response to treatment.

### Results

Fifty-five percent of children recovered and mean time to recovery was eight weeks while 13% ended as non-response to treatment. Independent predictors of longer time to recovery or non-response included low age, being admitted with WHZ <-3, no illness nor anaemia at admission, illness episodes during treatment, skipped or missed visits, low maternal age and not practising open defecation. Eighty-four percent of children had at least one and 59% at least two illness episodes during treatment. This increased treatment duration by 1 to 4 weeks. Thirty-five percent of children missed at least one treatment visit. One missed visit predicted 3 weeks longer and two or more missed visits 5 weeks longer treatment duration.

4274090 (https://zenodo.org/record/4274090#.YlXrS8jMI2w).

**Funding:** The data collection for this study was funded by Action Against Hunger France, European Commission's Civil Protection and Humanitarian aid Operations (grant number ECHO/-WF/BUD/2015/91065), Children's Investment Fund Foundation (grant: "CIFF03: Reinventing community based management of malnutrition"), European Commission's Civil Protection and Humanitarian aid Operations Enhanced Response Capacity (grant number ECHO/ERC/BUD/2016/91006) and Humanitarian Innovation Fund (HIF) a programme managed by ELRHA (Enhancing Learning and Research for Humanitarian Assistance). Funding provided by these organisations covered the study conception and field implementation of the study, including the salary of STK and VN for the full length of the study. Other authors received no specific funding for their work related to the current study. The funders had no role in study design, data collection and analysis, decision to publish, or preparation of the manuscript.

**Competing interests:** STK was previously employed by Nutriset, a producer of RUTF. HF has received research grants from ARLA Food for Health Centre, and also has research collaboration with Nutriset. Other authors declare no financial relationships with any organisations that might have an interest in the submitted work in the previous five years, and declare no other relationships or activities that could appear to have influenced the submitted work. The declared competing interests do not alter our adherence to PLOS ONE policies on sharing data and materials.

## Conclusions

Both longer time to recovery and higher non-response to treatment seem most strongly associated with illness episodes and missed visits during treatment. This indicates that prevention of illnesses would be key to shortening the treatment duration and that there is a need to seek ways to facilitate adherence.

## Introduction

Severe acute malnutrition (SAM) is a condition that occurs when the food intake does not meet the nutritional requirements either as a consequence of poor intake or disease [1]. In children 6–59 months of age, SAM is diagnosed when a child presents with a weight-for-height z-score (WHZ) <-3, a mid-upper-arm circumference (MUAC) <125 mm or nutritional oedema [2]. While the overall prevalence of SAM is unknown, in 2020 it was estimated that 2% of all children below the age of 5 years presented a WHZ <-3 translating to more than 13.6 million children suffering from severe wasting at any time [3]. Children with SAM have a 11.6 increased risk of mortality compared to children with no nutritional deficits living in the same contexts [4].

Generally SAM arises in contexts with social, political and economic factors affecting food availability and where infections and inflammation are common [1]. This is also why no single intervention has been shown to reduce the incidence that requires a holistic package of interventions [5].

According to the World Health Organisation (WHO), children with SAM without medical complications are treated as outpatients with weekly check-up visits [6]. The treatment includes a systematic antibiotic regimen and ready-to-use therapeutic foods (RUTF), prescribed according to the weight of the child and continued until discharge [6]. RUTF are energy and nutrient dense pastes usually composed of peanut butter, milk powder, oil, sugar and a vitamin and mineral complex designed to fulfil the nutritional needs of children recovering from SAM [7]. Recovery is defined as having reached a WHZ ≥-2 for those admitted with a WHZ <-3 or a MUAC ≥125 mm for those admitted with a MUAC <115 mm for at least two weeks [6]. Children that never attain recovery within a maximum treatment time are called non-response to treatment [8].

In 2020, around 5 million children were treated for SAM globally [9]. With 13.6 million children suffering from severe wasting at any time [3] and applying of incidence correction factor of 3.5 [10] to account for all new cases arising in a year, this translates to 47.6 million episodes of severe wasting in a year. Thus, the current coverage of treatment is around 10%. This is when excluding the burden of MUAC cases which, if added, would translate to an even lower coverage of treatment. Such low coverage warrants reflection on how to improve it possibly by optimising and better targeting treatment in order to expand it to more cases.

Children treated for SAM in different contexts present with varying mean treatment time and proportion of non-response to treatment [11–15]. Longer time in treatment and high percent of non-response to treatment increase the cost of treatment and question the effectiveness of the current treatment protocol. Little is known about the factors influencing treatment duration and non-response to treatment. Understanding who requires longer treatment time and which children are at the greatest risk of non-response to treatment could help guide the optimisation of SAM treatment and explain differences in program performance observed between different contexts.

The current study seeks to explore the predictors of time to recovery from and non-response to treatment of SAM in a community based treatment setting. The study is based on data collected during a randomised controlled trial comparing the efficacy of a reduced dose of RUTF with a standard dose on the treatment outcomes of children with uncomplicated SAM managed in outpatient care. The trial showed no significant effect of the dose reduction on the weight gain velocity, length of stay in treatment, recovery percent or proportion of non-response to treatment [16].

## Methods

### Ethics

The MANGO study was performed in accordance with the principles in the Declaration of Helsinki. The research protocol was approved by the national ethics committee (Comité d'éthique pour la Recherche en Santé) and the clinical trials board (Direction Générale de la Pharmacie, du Médicament et des Laboratoires) of Burkina Faso and was registered at the IRSCTN registry http://www.isrctn.com/ISRCTN50039021. Caregivers of participating children gave their informed consent in a written form.

### Study design and setting

This study is based on data collected prospectively as part of a randomized controlled trial (called MANGO), which compared the efficacy of a reduced RUTF dose to a standard RUTF dose in the management of SAM without medical complications in children 6–59 months of age in a non-inferiority design. The results from the randomized controlled trial have been published elsewhere [16–22]. The study recruited a total of 801 children which provides a good sample size for analysing predictors compared to previous studies looking at time to recovery with samples sizes starting from 200 and most around 400 children treated [23–34].

The trial was conducted in 10 health centres of the Fada N'Gourma health district in eastern Burkina Faso. Malaria was endemic with 69.3% of children in the region presenting a positive rapid test [35]. In 2016, the prevalence of severe wasting (WHZ <-3) and moderate wasting (WHZ between -3 and -2) was 2.4% and 8.6%, respectively [36]. Most households depend on small scale farming and livestock ownership [37] and 32% of the population lives more than 10 km away from nearest health post [38].

### Study participants and treatment protocol

The participants included all children enrolled in the clinical trial for which the enrolment procedures have been described in detail elsewhere [16]. In short, participants were children presenting with WHZ <-3 and/or MUAC <115 mm but without medical complications, at the 10 participating health centres. Children with any grade of oedema or no appetite were referred to inpatient care.

Treatment followed the Burkina Faso national community-based management of acute malnutrition (CMAM) guidelines [39] in all aspects except the RUTF dose. Half of the children received a reduced dose from the third treatment week onwards: One sachet per day to children weighing <7 kg and two to children ≥7 kg, representing a reduction between 30 and 54% compared to standard dose depending on the weight category [16]. Co-morbidities diagnosed during SAM treatment were managed according to national protocol [39]. In case of medical complications, weight loss of over 5% at any point or stagnant weight defined as no more than 100g weight gained over 4 weeks, during treatment, children were referred to

inpatient care, as per the Burkina national CMAM protocol [39]. Nutritional treatment was continued weekly until recovery.

Recovery was defined as having attained 1) a WHZ $\geq$-2 for those admitted with a WHZ <-3, or 2) a MUAC $\geq$125 mm for those admitted with a MUAC <115 mm, or 3) a WHZ $\geq$-2 and a MUAC $\geq$125 mm for those admitted with both WHZ<-3 and MUAC <115 mm, for 2 consecutive weeks and absence of illness. Children still not having reached the anthropometric recovery criteria by 16 weeks of treatment were declared as non-response to treatment. Other discharge categories included defaulting (defined as having missed 3 consecutive visits and confirmed alive), loss-to follow up (defined as having missed 3 consecutive visits and not confirmed alive), death, and false discharge (those discharged wrongly after verification of discharge anthropometrics).

## Data collection

Upon admission, the child's caregiver was interviewed regarding household socio-economic characteristics, care practices and recent morbidity of the child. Anthropometric measurements and a clinical examination were performed at each visit from admission to discharge. Weight was measured using an electronic scale (SECA 876) to the nearest 100 g, height using a wooden measuring board (locally made) to the nearest 1 mm, and MUAC using a non-stretchable colourless measuring tape to the nearest 1 mm. Z-scores were calculated using the WHO standards and STATA command zscore06 [40]. Haemoglobin (HemoCue) was measured at admission. All data were collected via tablets using the Open Data Kit (ODK1 software) and continuous data monitoring and cleaning was performed. Data monitoring included among other thing, checking duplicate entries and outliers, anthropometric decimal distributions and correct prescription of medication according to diagnosed conditions. Any potential data problem resulted in action. Data cleaning was based on double checks of electronic data against patient registries or therapeutic cards.

## Outcomes

Two outcomes were studied: 1) time to recovery defined as days passed from admission to treatment until discharge as recovered, and 2) non-response to treatment defined as not reaching the anthropometric recovery criteria within 16 weeks. Recovery and non-response were dichotomised to recovered or non-recovered and response or non-response, respectively. For the study of time to recovery, non-recovered cases were right censored contributing to the analysis of time to recovery until exit from study. Patients referred to inpatient care were excluded from the analysis in order to limit potential bias that could be introduced due to their short length of stay in treatment.

## Predictors

Predictors included in the analyses were variables describing 1) admission characteristics, 2) morbidity and compliance to treatment visits during treatment and 3) household characteristics. Admission characteristics included sex, age, WHZ, MUAC, height for-age z-score (HAZ), admission criteria (WHZ <-3, MUAC <115 mm or both), any illness, anaemia, breastfeeding status and low birth weight. Illness at admission was defined as any caregiver reported illness (including cough, diarrhoea, fever, vomiting, skin lesions) observed in the week prior to admission or diagnosed by study nurse upon admission. Anaemia was defined as a haemoglobin level <110 g/l [41]. Low birth weight (<2500 g) was confirmed from an official birth certificate or health card. Morbidity and compliance variables included an episode of malaria, acute respiratory illness (ARI) or diarrhoea during treatment and number of illness episodes as well

as number of skipped and missed visits. Malaria episode during treatment was defined as an armpit temperature of >37.5˚C, a positive malaria rapid diagnostic test (RDT) and a negative RDT at admission. ARI was defined as cough reported by caregiver in the past week or diagnosed by study nurse during visit. Diarrhoea included acute, persistent or dysenteric forms and was defined as 3 or more loose stools per day as reported by caregiver in the past week or diagnosed by study nurse. Illness episodes included any caregiver reported or nurse diagnosed illness in the past week. Skipped visits were those that were planned in advance and thus the caregiver was prescribed double dose of RUTF to cover 2 weeks of home treatment. Missed visits were unplanned and thus represent gaps in RUTF prescription. Household characteristics included caregiver's age, education level, whether caregiver was the household head, number of children under 5 years of age in the household, water source, open defecation (the practice of defecating in nature instead of a toilet facility) or not, food security status, distance from health centre and urban or rural setting. Safe water source was defined as a protected well, pump or tap while unsafe water source included unprotected wells and rivers, lakes and ponds. Food security assessment was based on the Household Food Insecurity Access Scale (HFIAS) [42]. Distance from health centre was estimated by the caregivers as the time needed for a return trip with the available transportation means.

## Data analysis

All children included in the MANGO trial regardless of their treatment arm (reduced RUTF or standard RUTF) were included in the analysis. Baseline characteristics of the study population are summarized as percent (n) or mean ±SD. Cox proportional hazard regression models were fitted to quantify effects of predictors on time to recovery with resulting hazard ratios describing the increased or decreased chance of recovery. Kaplan Meier plots were used to visualize results concerning age and admission categories. Restricted mean time to recovery was estimated for significant categorical predictors. Logistic regression was used to evaluate predictors of non-response to treatment. Both unadjusted and models adjusted for sex and age at admission were fitted. A p-value below 0.05 was used to declare statistical significance. All analyses were performed using STATA 15 (StataCorp, USA) and Kaplan Meier plots were produced in GraphPad Prism 8 (GraphPad Software, USA).

## Results

In total, 801 children were enrolled in the RCT, of which 54% (n = 433) recovered with a median [IQR] time to recovery of 8 weeks [5–12]. Thirteen percent (n = 101) were considered non-response to treatment, 20% (n = 157) were referred to inpatient care due to stagnant weight, weight loss or medical complication, 10% (n = 83) defaulted, 3% (n = 24) were falsely discharged, 2 children died and 1 was lost to follow up. Excluding the 157 referrals, 644 children contributed to the analysis of time to recovery and all 801 to the analysis of non-response to treatment.

At admission, children were on average 13 months old, 86% were breastfed, 79% had an illness, and 80% had anaemia (Table 1). Most households (83%) had access to a safe drinking

**Table 1. Characteristics of children included in the analysis of time to recovery.**

| Characteristics | Values |
|---|---|
| 1. Admission characteristics | |
| Male, % (n) | 47 (305) |
| Age, months | 13.4 ± 8.4 |

*(Continued)*

**Table 1.** (Continued)

| Characteristics | Values |
|---|---|
| WHZ at admission | -3.0 ± 0.7 |
| MUAC at admission, mm | 113.1 ± 6.2 |
| HAZ at admission | -2.4 ± 1.3 |
| Admission criteria, % (n) | |
| WHZ only | 27 (171) |
| MUAC only | 39 (252) |
| both WHZ and MUAC | 34 (221) |
| Illness, % (n) | 79 (507) |
| Anaemia, % (n) | 80 (514) |
| Low birth weight, % (n)[1] | 20 (84) |
| Breastfeeding, % (n) | 86 (552) |
| 2. Morbidity and compliance during treatment | |
| Malaria episode, % (n) | 16 (105) |
| ARI episode, % (n) | 47 (303) |
| Diarrhoea episode, % (n) | 27 (177) |
| Number of illness episodes, % (n) | |
| none | 16 (102) |
| one | 25 (163) |
| two | 18 (113) |
| three or more | 41 (266) |
| Number of skipped* visits, % (n) | |
| none | 48 (309) |
| one | 25 (163) |
| two or more | 27 (172) |
| Number of missed* visits, % (n) | |
| none | 65 (418) |
| one | 19 (120) |
| two or more | 16 (106) |
| 3. Household characteristics | |
| Maternal age, years | 27.9 ± 7.7 |
| Mother has some formal education, % (n) | 24 (154) |
| Caregiver is the household head, % (n) | 3 (21) |
| Number of children under 5 in the household, % (n) [1] | |
| only index child | 31 (201) |
| two | 34 (220) |
| three or more | 34 (218) |
| Using safe water source, % (n) | 83 (533) |
| Open defecation, % (n) | 77 (496) |
| Household is food secure, % (n) | 89 (575) |
| >30 min return distance from health centre, % (n) | 63 (407) |
| Urban setting, % (n) | 14 (89) |

Values are mean ± SD unless otherwise indicated

[1] Birth weight was only available for 410 children and number of children in the household for 639 of all 644 children included in the time to recovery analysis

* skipped visits refer to those that were planned and thus benefitted from double RUTF prescription as opposed to missed visits that were unplanned

ARI, acute respiratory infection; HAZ, height-for-age z-score; MUAC, mid-upper arm circumference; WHZ, weight-for-height z-score.

water source but 79% were practising open defecation. During treatment, 84% of children had at least one illness episode with 59% reporting at least two episodes. Up to 35% of children missed at least one treatment visit with 16% missing more than one visit.

Unadjusted and adjusted estimates of associations with time to recovery are presented in Table 2. When adjusted for sex and age at admission, independent predictors of longer time to recovery included low age, low WHZ at admission, being admitted with WHZ <-3 (compared to only MUAC <115mm), not having illness or anaemia at admission, having any co-morbidity episode during treatment, higher number of missed treatment visits, low maternal age and not practising open defecation. For example, for every 1 z-score increase in WHZ at admission, children have 26% higher chance of recovery.

The mean time to recovery for significant categorical variables are presented in Table 3. Children <12 months of age required 13 more days to reach recovery compared to children ≥12 months of age (Fig 1). Being admitted with only MUAC criteria and thus a WHZ ≥-3 predicted faster recovery compared to those admitted with WHZ only (Fig 2) with an average of 15 days shorter time to recovery compared to WHZ only or 10 days shorter compared to those with both admission criteria (Table 3). Illness and malaria at admission were associated with 6 and 14 days faster recovery, respectively. On the contrary, having one, two or more than two illness episodes during treatment was associated with 1, 4 and 8 weeks longer time to recovery, respectively, compared to those with no illness episodes during treatment. One skipped visit increased time to recovery by 2 weeks and one missed visit by 3 weeks. Open defecation was associated with 1 week faster recovery. Including children referred to inpatient care in the analysis of different factors with time to recovery resulted in similar associations.

Factors associated with non-response to treatment were largely similar to those associated with time to recovery in that factors predicting a slower recovery also predicted non-response to treatment and factors predicting faster recovery also predicted not ending up non-response (see Table 4).

## Discussion

In this study, only 55% of children admitted to outpatient treatment of SAM recovered while 13% were discharged non-response to treatment. The mean time to recovery was 8 weeks and was most strongly associated with illness episodes and missed and skipped visits during treatment. Up to 59% of children had at least two illness episodes during treatment which increased the treatment duration by nearly 4 weeks with two episodes and over 7 weeks with more than two episodes. Up to 35% of children missed at least one visit which increased the time to recovery by 3 weeks with one and over 5 weeks with two or more missed visits. Similarly, non-response to treatment seemed most strongly associated with illness episodes and missed visits during treatment increasing the odds of non-response to 1.95 for each additional illness episode and 2.04 for each additional missed visit.

The mean time to recovery was longer than reported by most previous studies conducted in outpatient settings [14, 15, 27, 31, 32, 34, 43, 44]. Most of the variability in length of stay in treatment between different studies and contexts can probably be ascribed to differences in discharge criteria: in many studies recovery was declared when children reached a weight-for-height >85% of WHO median growth reference regardless of being admitted with low WHZ or MUAC [14, 24–26, 30–32, 43, 45]. Additionally, often the recommendation [6] of presenting the anthropometric recovery criteria for at least 2 weeks is not followed [14, 24–26, 30, 44]. These deviations from the WHO issued protocol have consequences on the program performance indicators [46].

**Table 2. Predictors of time to recovery (days) among 644 children during outpatient treatment of severe acute malnutrition.**

| Predictors | n | | Unadjusted | | | Sex and age adjusted[2] | | |
|---|---|---|---|---|---|---|---|---|
| | Event | Censored | HR[1] | 95% CI | p-value | HR[1] | 95% CI | p-value |
| 1.Admission characteristics | | | | | | | | |
| Sex | | | | | | | | |
| Male | 209 | 96 | Ref | | | Ref | | |
| Female | 224 | 115 | 0.86 | 0.71; 1.04 | 0.12 | 0.88 | 0.73; 1.07 | 0.20 |
| Age, months | 433 | 211 | 1.01 | 1.00; 1.03 | 0.004 | 1.01 | 1.00; 1.02 | 0.007 |
| WHZ | 433 | 211 | 1.12 | 0.98; 1.27 | 0.10 | 1.26 | 1.09; 1.46 | 0.002 |
| MUAC, mm | 433 | 211 | 1.01 | 1.00; 1.03 | 0.15 | 1.00 | 0.99; 1.02 | 0.85 |
| HAZ | 433 | 211 | 0.93 | 0.87; 1.00 | 0.055 | 0.95 | 0.88; 1.02 | 0.16 |
| Admission criteria | | | | | | | | |
| WHZ only | 102 | 69 | Ref | | | Ref | | |
| MUAC only | 185 | 67 | 1.31 | 1.03; 1.67 | 0.028 | 1.80 | 1.36; 2.38 | <0.001 |
| both WHZ & MUAC | 146 | 75 | 1.05 | 0.82; 1.36 | 0.68 | 1.26 | 0.97; 1.65 | 0.086 |
| Any illness | | | | | | | | |
| No | 78 | 59 | Ref | | | Ref | | |
| Yes | 355 | 152 | 1.40 | 1.09; 1.79 | 0.008 | 1.35 | 1.05; 1.73 | 0.017 |
| Anaemia | | | | | | | | |
| No | 78 | 52 | Ref | | | Ref | | |
| Yes | 355 | 159 | 1.50 | 1.18; 1.92 | 0.001 | 1.60 | 1.25; 2.06 | <0.001 |
| Low birth weight | | | | | | | | |
| No | 222 | 104 | Ref | | | Ref | | |
| Yes | 54 | 30 | 0.81 | 0.60; 1.09 | 0.16 | 0.80 | 0.60; 1.08 | 0.15 |
| Breastfeeding | | | | | | | | |
| No | 66 | 26 | Ref | | | Ref | | |
| Yes | 367 | 185 | 0.64 | 0.49; 0.84 | 0.001 | 0.70 | 0.48; 1.01 | 0.057 |
| 2.Morbidity and compliance during treatment | | | | | | | | |
| Malaria episode | | | | | | | | |
| No | 382 | 157 | Ref | | | Ref | | |
| Yes | 51 | 54 | 0.42 | 0.32; 0.57 | <0.001 | 0.43 | 0.32; 0.58 | <0.001 |
| ARI episode | | | | | | | | |
| No | 248 | 93 | Ref | | | Ref | | |
| Yes | 185 | 118 | 0.44 | 0.37; 0.54 | <0.001 | 0.45 | 0.37; 0.55 | <0.001 |
| Diarrhoea episode | | | | | | | | |
| No | 342 | 125 | Ref | | | Ref | | |
| Yes | 91 | 86 | 0.43 | | | 0.44 | 0.35; 0.56 | <0.001 |
| Number of illness episodes | 433 | 211 | 0.57 | 0.53; 0.61 | <0.001 | 0.57 | 0.53; 0.61 | <0.001 |
| Number of skipped* visits | 433 | 211 | 0.67 | 0.61; 0.73 | <0.001 | 0.66 | 0.60; 0.73 | <0.001 |
| Number of missed* visits | 433 | 211 | 0.60 | 0.53; 0.67 | <0.001 | 0.59 | 0.52; 0.67 | <0.001 |
| 3.Household characteristics | | | | | | | | |
| Maternal age, years | 433 | 211 | 1.02 | 1.00; 1.03 | 0.029 | 1.01 | 1.00; 1.03 | 0.048 |
| Maternal education | | | | | | | | |
| No formal education | 333 | 157 | Ref | | | Ref | | |
| Some formal education | 100 | 54 | 0.84 | 0.67; 1.05 | 0.13 | 0.84 | 0.67; 1.05 | 0.13 |
| Number of children under 5 in the household | 432 | 207 | 1.04 | 0.97; 1.11 | 0.31 | 1.05 | 0.98; 1.12 | 0.16 |
| Caregiver is the household head | | | | | | | | |
| No | 416 | 207 | Ref | | | Ref | | |
| Yes | 17 | 4 | 1.51 | 0.93; 2.45 | 0.098 | 1.36 | 0.83; 2.23 | 0.22 |

*(Continued)*

**Table 2.** (Continued)

| Predictors | n | | Unadjusted | | | Sex and age adjusted[2] | | |
|---|---|---|---|---|---|---|---|---|
| | Event | Censored | HR[1] | 95% CI | p-value | HR[1] | 95% CI | p-value |
| Safe water source | | | | | | | | |
| No | 62 | 49 | Ref | | | Ref | | |
| Yes | 371 | 162 | 1.27 | 0.97; 1.66 | 0.082 | 1.24 | 0.95; 1.62 | 0.12 |
| Open defecation | | | | | | | | |
| No | 90 | 58 | Ref | | | Ref | | |
| Yes | 343 | 153 | 1.30 | 1.03; 1.65 | 0.025 | 1.32 | 1.04; 1.67 | 0.021 |
| Food insecure household | | | | | | | | |
| No | 383 | 192 | Ref | | | Ref | | |
| Yes | 50 | 19 | 0.92 | 0.69; 1.24 | 0.59 | 0.86 | 0.64; 1.16 | 0.33 |
| Return time from health centre | | | | | | | | |
| ≤30 min | 156 | 81 | Ref | | | Ref | | |
| >30 min | 277 | 130 | 0.97 | 0.80; 1.19 | 0.80 | 1.01 | 0.83; 1.23 | 0.93 |
| Setting | | | | | | | | |
| Rural | 375 | 180 | Ref | | | Ref | | |
| Urban | 58 | 31 | 0.92 | 0.70; 1.22 | 0.58 | 0.89 | 0.67; 1.17 | 0.40 |

[1] HR>1 means faster recovery

[2] when analysing sex as a predictor, only age was included as adjustment and when analysing age as a predictor only sex was included as adjustment

* skipped visits refer to those that were planned and thus benefitted from double RUTF prescription as opposed to missed visits that were unplanned

ARI, acute respiratory infection; HR, Hazard Ratio; MUAC, mid-upper arm circumference; WHZ, weight-for-height z-score.

Illness episodes during treatment were the strongest predictor of recovery and non-response. Few outpatient studies have looked at co-morbidities occurring during treatment. Yebyo et al. (2013) found a significant association between co-morbidities and lower recovery rate but did not distinguish between illnesses diagnosed at admission or during treatment [14]. The relatively long treatment duration in the current study probably contributed to the observation that illnesses diagnosed during treatment were more predictive of recovery than those diagnosed at admission. Children were systematically treated for the diagnosed illnesses and the observation that these episodes were still associated with remarkably longer treatment time is noteworthy. It highlights the importance of preventing the conditions as it seems that once children catch diarrhoea, malaria or ARI during SAM treatment, the illnesses substantially slow their recovery trajectory.

One missed visit increased the time to recovery by more than 3 weeks. The reason for missing a visit was in most cases related to time or accessibility impediments to travel to the health centre to receive treatment. This calls for seeking more flexible and possibly more closely available services; flexible in the sense of providing services on several week days instead of just one and more closely available potentially in the form of community health worker delivered treatment [47]. Interestingly, even when children had benefitted from double RUTF prescription, one skipped visit still increased the time to recovery by more than 2 weeks. This suggests that increased visit spacing could result in longer treatment time. This said, part of the increased time (1 week) can be explained by missed and skipped visits occurring towards the end of treatment artificially lengthening treatment time when recovery can only be declared upon reaching discharge criteria during two consecutive visits.

We observed that higher age was associated with faster recovery. Previous studies in outpatient settings have found positive [27, 34], negative [32] and no [14, 31, 33, 43] associations

**Table 3. Estimated restricted mean time (days) to recovery from SAM by significant predictor characteristics.**

| Predictors | mean (days) | Unadjusted | | | Sex and age adjusted[2] | | |
|---|---|---|---|---|---|---|---|
| | | Difference | 95%CI | p-value | Difference | 95%CI | p-value |
| 1.Admission characteristics | | | | | | | |
| Age | | | | | | | |
| <12 months | 75.7 | | | | | | |
| ≥12 months | 62.3 | -13.5 | -18.9; -8.1 | <0.001 | -13.4 | -18.8; -8.0 | <0.001 |
| Admission criteria | | | | | | | |
| WHZ only | 73.8 | | | | | | |
| MUAC only | 66.0 | -7.8 | -14.8; -0.9 | 0.027 | -15.1 | -22.6; -7.6 | <0.001 |
| WHZ & MUAC | 73.5 | -0.2 | -7.4; 6.9 | 0.95 | -4.8 | -12.1; 2.5 | 0.20 |
| Any illness | | | | | | | |
| No | 76.5 | | | | | | |
| Yes | 68.9 | -7.6 | -14.4; -0.9 | 0.026 | -6.2 | -12.9; 0.6 | 0.075 |
| Anaemia | | | | | | | |
| No | 81.4 | | | | | | |
| Yes | 67.7 | -13.7 | -20.1; -7.4 | <0.001 | -14.5 | -21.1; -7.9 | <0.001 |
| Breastfeeding | | | | | | | |
| No | 57.8 | | | | | | |
| yes | 72.6 | 14.7 | 7.1; 22.4 | <0.001 | 8.9 | -1.3; 19.1 | 0.086 |
| 2.Morbidity and compliance during treatment | | | | | | | |
| Malaria episode | | | | | | | |
| no | 66.2 | | | | | | |
| yes | 91.5 | 25.3 | 19.2; 31.4 | <0.001 | 24.2 | 18.0; 30.3 | <0.001 |
| ARI episode | | | | | | | |
| no | 58.0 | | | | | | |
| yes | 83.0 | 24.9 | 20.0; 29.9 | <0.001 | 24.0 | 19.0; 29.1 | <0.001 |
| Diarrhoea episode | | | | | | | |
| no | 63.6 | | | | | | |
| yes | 87.9 | 24.4 | 19.0; 29.7 | <0.001 | 23.0 | 17.5; 28.4 | <0.001 |
| Number of illness episodes | | | | | | | |
| none | 39.2 | | | | | | |
| one | 46.6 | 7.4 | 1.6; 13.1 | 0.012 | 6.9 | 1.2; 12.6 | 0.017 |
| two | 66.5 | 27.3 | 21.0; 33.7 | <0.001 | 26.9 | 20.6; 33.2 | <0.001 |
| more than two | 92.2 | 53.0 | 47.8; 58.2 | <0.001 | 52.7 | 47.4; 58.0 | <0.001 |
| Number of skipped[1] visits | | | | | | | |
| none | 56.1 | | | | | | |
| one | 71.4 | 15.3 | 9.1; 21.5 | <0.001 | 16.2 | 10.2; 22.2 | <0.001 |
| more than one | 91.1 | 35.0 | 29.6; 40.3 | <0.001 | 35.6 | 30.2; 41.0 | <0.001 |
| Number of missed[1] visits | | | | | | | |
| none | 59.1 | | | | | | |
| one | 83.0 | 23.9 | 17.8; 30.1 | <0.001 | 23.2 | 16.8; 29.6 | <0.001 |
| more than one | 98.7 | 39.6 | 34.5; 44.7 | <0.001 | 39.4 | 34.1; 44.6 | <0.001 |
| 3.Household characteristics | | | | | | | |
| Open defecation | | | | | | | |
| no | 75.7 | | | | | | |

(*Continued*)

**Table 3.** (Continued)

| Predictors | mean (days) | Unadjusted | | | Sex and age adjusted[2] | | |
|---|---|---|---|---|---|---|---|
| | | Difference | 95%CI | p-value | Difference | 95%CI | p-value |
| yes | 68.9 | -6.7 | -13.2; -0.2 | 0.042 | -7.0 | -13.5; -0.5 | 0.036 |

[1] skipped visits refer to those that were planned and thus benefitted from double RUTF prescription as opposed to missed visits that were unplanned

[2] when analysing age categories as a predictor only sex was included as adjustment

ARI, acute respiratory infection; MUAC, mid-upper arm circumference; OR, odds ratio; WHZ, weight for height z-score.

between age and recovery. This could partly be due to the use of different cut-offs for defining age groups ranging from 12 months to 36 months. Worth noting though, in our study only 10% of children were ≥24 months old at admission, meaning that in a context with an older SAM population, the association could be different.

Being admitted based on MUAC only (ie. MUAC <115 mm and WHZ ≥-3) was associated with 2 weeks quicker recovery and 0.41 the odds of non-response than being admitted with WHZ only (ie. WHZ <-3 and MUAC ≥115 mm). This indicates either that children presenting with a low WHZ potentially have a different pathophysiological status that requires a longer treatment time or that reaching a WHZ ≥-2 takes longer than reaching a MUAC ≥125

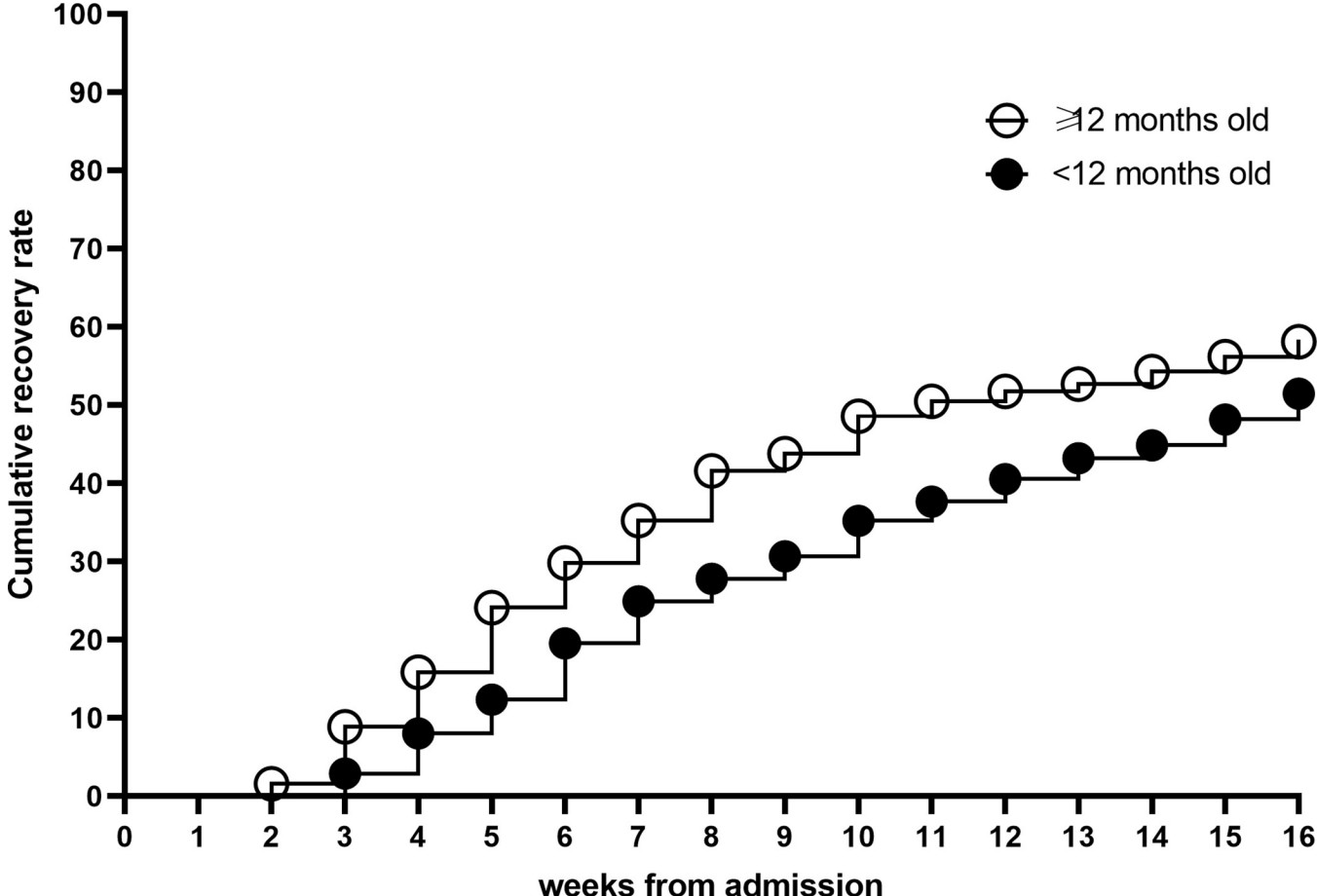

**Fig 1. Kaplan Meier plot of cumulative recovery from SAM by age category during outpatient treatment.**

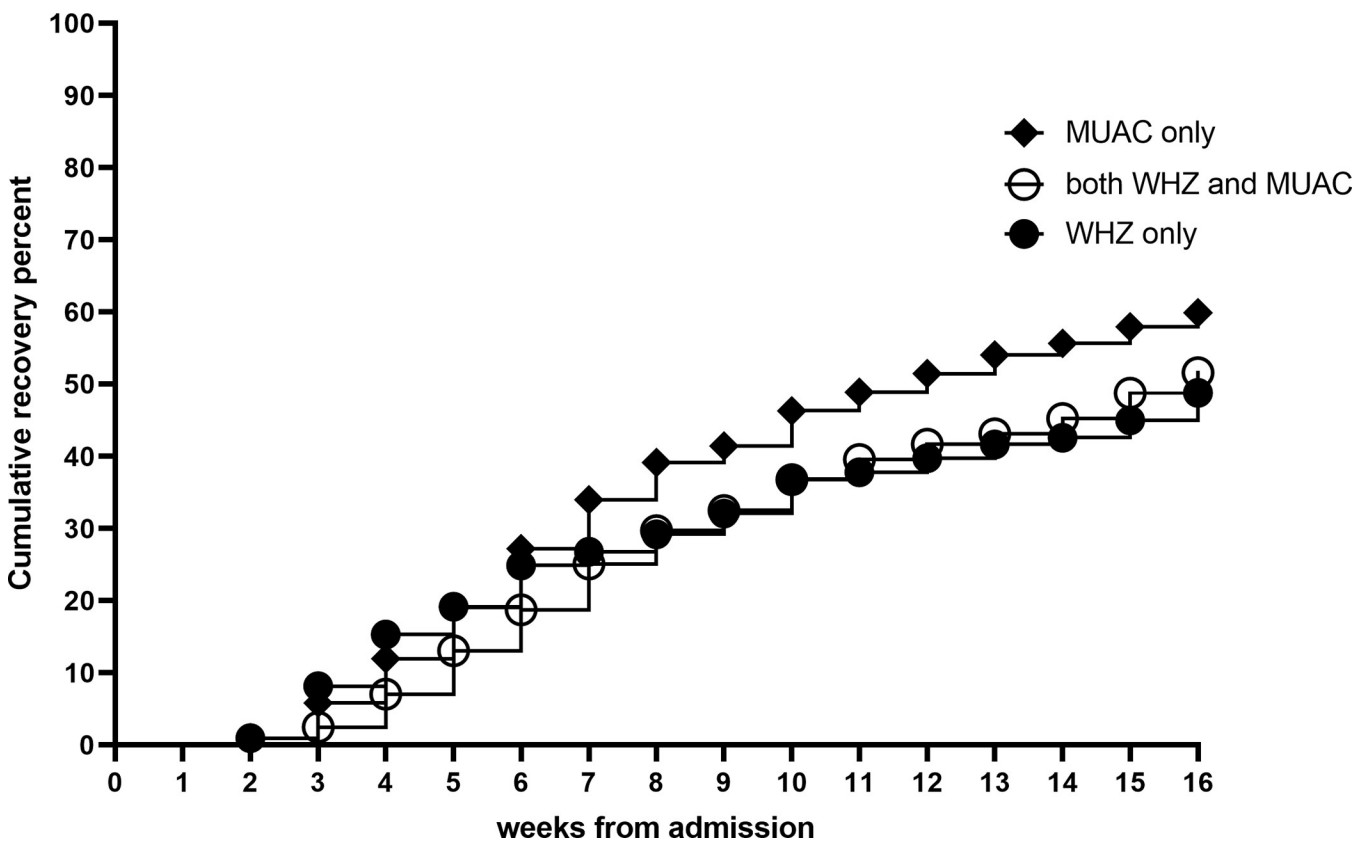

**Fig 2. Kaplan Meier plot of cumulative recovery from SAM by admission criteria during outpatient treatment.**

mm. This observation is to be considered when observing success rates from programs implementing MUAC only admission criteria as based on the current study excluding children with WHZ <-3 but MUAC > 115mm would result in better recovery rate.

Illness at admission were associated with faster recovery, contrary to results from most previous studies showing slower recovery in inpatient [24, 25, 28–30, 45, 48] and outpatient settings [14] among those admitted with co-morbidity. Interestingly however, one study from Gambia showed that higher cortisol at admission (indicating acute illness) predicted higher WAZ gain during treatment [49]. The authors suspected that this was due to children with high cortisol being more sick and when treated for their condition responding faster by also gaining weight. In somewhat similar lines, another study found that children that failed the appetite test at admission, possibly indicating sub-clinical illness, had higher weight gains during treatment compared to children having passed the appetite test [50]. Our observation offers support to this hypothesis in that it would seem that children with co-morbidities respond fastest to the treatment. It could be that their malnutrition is a secondary condition related to a primary co-morbidity that when managed correctly allows a rapid return to a normal health and nutrition status. Consequently, children with no apparent co-morbidities at admission possibly have a different causal pathway to malnutrition and maybe a different pathophysiological state leading to slower response to treatment. It is also possible that they present with some underlying chronic hard-to-detect pathology such as environmental enteropathy [51–53] or congenital heart diseases [54] that increase the nutrient needs and could affect the time to recovery.

**Table 4. Predictors of non-response to outpatient treatment of SAM in 801 patients without medical complications at admission.**

| Predictors | n | | Unadjusted | | | Sex and age adjusted[2] | | |
|---|---|---|---|---|---|---|---|---|
| | Non-response | Response | OR | 95% CI | p-value | OR | 95% CI | p-value |
| 1.Admission characteristics | | | | | | | | |
| Sex | | | | | | | | |
| Male | 41 | 355 | Ref | | | Ref | | |
| Female | 60 | 345 | 1.51 | 0.99; 2.30 | 0.058 | 1.46 | 0.95; 2.24 | 0.082 |
| Age | 101 | 700 | 0.97 | 0.94; 1.00 | 0.039 | 0.97 | 0.94; 1.00 | 0.051 |
| WHZ | 101 | 700 | 0.96 | 0.72; 1.30 | 0.80 | 0.80 | 0.58; 1.09 | 0.16 |
| MUAC | 101 | 700 | 0.98 | 0.95; 1.01 | 0.30 | 1.00 | 0.96; 1.04 | 0.98 |
| HAZ | 101 | 700 | 1.11 | 0.94; 1.30 | 0.22 | 1.05 | 0.89; 1.24 | 0.58 |
| Admission criteria | | | | | | | | |
| WHZ only | 30 | 179 | Ref | | | Ref | | |
| MUAC only | 33 | 276 | 0.71 | 0.42; 1.21 | 0.21 | 0.41 | 0.22; 0.75 | 0.004 |
| WHZ & MUAC | 38 | 245 | 0.93 | 0.55; 1.55 | 0.77 | 0.66 | 0.38; 1.14 | 0.14 |
| Any illness | | | | | | | | |
| no | 31 | 143 | Ref | | | Ref | | |
| yes | 70 | 557 | 0.58 | 0.37; 0.92 | 0.020 | 0.61 | 0.38; 0.97 | 0.038 |
| Anaemia | | | | | | | | |
| no | 29 | 144 | Ref | | | Ref | | |
| yes | 72 | 556 | 0.64 | 0.40; 1.03 | 0.065 | 0.63 | 0.39; 1.01 | 0.057 |
| Low birth weight | | | | | | | | |
| no | 54 | 338 | Ref | | | Ref | | |
| yes | 19 | 91 | 1.31 | 0.74; 2.31 | 0.34 | 1.25 | 0.70; 2.23 | 0.44 |
| Breastfeeding | | | | | | | | |
| no | 9 | 103 | Ref | | | Ref | | |
| yes | 92 | 597 | 1.76 | 0.86; 3.61 | 0.12 | 1.19 | 0.48; 2.94 | 0.71 |
| 2.Morbidity and compliance during treatment | | | | | | | | |
| Malaria episode | | | | | | | | |
| no | 65 | 603 | Ref | | | Ref | | |
| yes | 36 | 97 | 3.44 | 2.17; 5.46 | <0.001 | 3.41 | 2.14; 5.42 | <0.001 |
| ARI episode | | | | | | | | |
| no | 23 | 380 | Ref | | | Ref | | |
| yes | 78 | 320 | 4.03 | 2.47; 6.56 | <0.001 | 3.81 | 2.33; 6.23 | <0.001 |
| Diarrhoea episode | | | | | | | | |
| no | 50 | 507 | Ref | | | Ref | | |
| yes | 51 | 193 | 2.68 | 1.75; 4.09 | <0.001 | 2.57 | 1.68; 3.95 | <0.001 |
| Number of illness episodes | 101 | 700 | 1.95 | 1.72; 2.21 | <0.001 | 1.95 | 1.72; 2.21 | <0.001 |
| Number of skipped[1] visits | 101 | 700 | 1.95 | 1.65; 2.29 | <0.001 | 1.94 | 1.65; 2.29 | <0.001 |
| Number of missed[1] visits | 101 | 700 | 2.01 | 1.69; 2.38 | <0.001 | 2.04 | 1.71; 2.42 | <0.001 |
| 3.Household characteristics | | | | | | | | |
| Maternal age | 101 | 700 | 1.00 | 0.97; 1.02 | 0.74 | 1.00 | 0.97; 1.03 | 0.92 |
| Mother has received some formal education | | | | | | | | |
| no | 70 | 536 | Ref | | | Ref | | |
| yes | 31 | 164 | 1.45 | 0.92; 2.29 | 0.113 | 1.45 | 0.91; 2.29 | 0.12 |
| Number of children under 5 in the household | 101 | 700 | 0.86 | 0.72; 1.03 | 0.103 | 0.83 | 0.69; 1.00 | 0.054 |
| Safe water source | | | | | | | | |
| no | 18 | 122 | Ref | | | Ref | | |
| yes | 83 | 578 | 0.97 | 0.56; 1.68 | 0.92 | 1.06 | 0.61; 1.84 | 0.83 |

(*Continued*)

**Table 4.** (Continued)

| Predictors | n | | Unadjusted | | | Sex and age adjusted[2] | | |
|---|---|---|---|---|---|---|---|---|
| | Non-response | Response | OR | 95% CI | p-value | OR | 95% CI | p-value |
| Open defecation | | | | | | | | |
| no | 34 | 158 | Ref | | | Ref | | |
| yes | 67 | 542 | 0.57 | 0.37; 0.90 | 0.016 | 0.58 | 0.37; 0.91 | 0.018 |
| Food insecure household | | | | | | | | |
| no | 90 | 613 | Ref | | | Ref | | |
| yes | 11 | 87 | 0.86 | 0.44; 1.67 | 0.66 | 0.97 | 0.50; 1.91 | 0.94 |
| Return time from health centre | | | | | | | | |
| ≤30min | 41 | 263 | Ref | | | Ref | | |
| >30min | 60 | 437 | 0.88 | 0.58; 1.35 | 0.56 | 0.82 | 0.53; 1.27 | 0.37 |
| Setting | | | | | | | | |
| rural | 80 | 608 | Ref | | | Ref | | |
| urban | 21 | 92 | 1.73 | 1.02; 2.94 | 0.041 | 1.93 | 1.13; 3.31 | 0.017 |

[1] skipped visits refer to those that were planned and thus benefitted from double RUTF prescription as opposed to missed visits that were unplanned

[2] when analysing sex as a predictor, only age was included as adjustment and when analysing age as a predictor only sex was included as adjustment

ARI, acute respiratory infection; MUAC, mid-upper arm circumference; OR, odds ratio; WHZ, weight-for-height z-score.

Anaemia at admission was associated with faster recovery. This is in contrast to previous studies in inpatient settings reporting slower recovery rates among children with anaemia at admission [24, 25, 30, 45]. It could be that children admitted to inpatient care because of medical complications and in addition presenting with anaemia have a different pathophysiologic profile to children in outpatient care with anaemia but who don't present medical complications and that would explain a slower response to treatment. It could also be that the inpatient treatment with therapeutic feeds such as F75 and F100 that do not contain iron [55] would slow down the recovery of anaemic patients. Regardless, anaemia is a condition driven by multiple factors including infections and nutritional deficiencies [56]. Similarly to the possible explanation for the faster recovery among children with illnesses at admission, it could be that these children, when managed properly via medical and nutritional treatment, respond quickly to treatment.

Open defecation was associated with slightly faster recovery. This association was contrary to what we expected. It could be that due to poor hygiene conditions these children entered treatment after an enteric illness and weight loss and subsequently responded fast to treatment. However, controlling for diarrhoea at admission did not reduce the association.

Living in a more urban setting was associated with higher non-response rate compared to a more rural setting and we do not have a hypothesis for why this should be. Among previous studies looking into factors influencing time to recovery two have reported no association between residence and recovery [31, 34] and one reported a quicker recovery among those coming from a more urban setting [27].

We found no association between HAZ at admission and time to recovery or non-response to treatment. Categorising HAZ into <-2 and ≥-2 did not reveal an association either. This is worth noting as there has been some interest in looking at the treatment outcomes of concurrently stunted and wasted children. Previously there has even been concern that short wasted children would not respond adequately to treatment although this has been shown not to be the case [57]. Our findings are consistent with this observation.

This study has several strengths including that it was done prospectively as part of a clinical trial with few missing data. The data quality was high with nurse diagnosed morbidity and

strict respect of recovery criteria to investigate time to recovery and non-response to treatment.

The study also has limitations. The population studied was very young (mostly <24 months old) so some of the associations could be different if the full age range of 6–59 months were present. Also, the use of strict referral criteria identifying children with stagnant weight gain and weight loss led to high referral rate (20%) and many children being excluded from the time to recovery analysis. However, including these children in the analysis did not change the associations. Additionally, the findings could be context specific as the causal pathways and the response to treatment may be different in other settings.

In conclusion, illness episodes during treatment were common and also the strongest predictor of non-recovery and non-response, in addition to missed visits. Over half of the children suffered at least 2 illness episodes during treatment increasing time to recovery by nearly 4 weeks with two episodes and over 7 weeks with more than two episodes. This indicates that while correct diagnosis and management of co-morbidities is crucial, prevention would be key to shortening the treatment duration. Missing a visit during treatment also increased treatment time considerably by 3 weeks which calls for reflecting the service delivery to better accommodate clients' needs to rearrange appointments and potentially further decentralise health services. In general, we call for more data on children with SAM, both prior and during treatment, in order to better understand causal pathways and pathophysiology of children admitted to care and subsequently their specific needs for recovery.

## Acknowledgments

We thank the study participants and caregivers for participating in the trial and the research teams for all their efforts in implementing the trial. Any views or opinions reflected here are those of the authors, expressed in a personal capacity, and do not necessarily reflect those of their employer.

## Author Contributions

**Conceptualization:** Suvi T. Kangas.

**Data curation:** Suvi T. Kangas, Victor Nikièma.

**Formal analysis:** Suvi T. Kangas.

**Funding acquisition:** Cécile Salpéteur.

**Methodology:** Suvi T. Kangas, Christian Ritz.

**Project administration:** Suvi T. Kangas, Cécile Salpéteur, Victor Nikièma.

**Resources:** Suvi T. Kangas, Victor Nikièma.

**Supervision:** Henrik Friis, Pernille Kaestel.

**Validation:** Christian Ritz, Henrik Friis, André Briend, Pernille Kaestel.

**Writing – original draft:** Suvi T. Kangas.

**Writing – review & editing:** Suvi T. Kangas, Cécile Salpéteur, Victor Nikièma, Christian Ritz, Henrik Friis, André Briend, Pernille Kaestel.

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
