## [Decision Letter · Decision Letter 0]

19 Apr 2021

PONE-D-20-35912

Predictors of time to recovery and non-response during outpatient treatment of severe acute malnutrition

PLOS ONE

Dear Dr. Kangas,

Thank you for submitting your manuscript to PLOS ONE. After careful consideration, we feel that it has merit but does not fully meet PLOS ONE’s publication criteria as it currently stands. Therefore, we invite you to submit a revised version of the manuscript that addresses the points raised during the review process.

We look forward to receiving your revised manuscript.

Kind regards,

Seth Adu-Afarwuah

Academic Editor

PLOS ONE

Journal Requirements:

'STK was previously employed by Nutriset, a producer of RUTF. HF has received research grants from ARLA Food for Health Centre, and also has research collaboration with Nutriset. Other authors declare no financial relationships

with any organisations that might have an interest in the submitted work in the previous five years, and declare no other relationships or activities that could appear to have influenced the submitted work.'

a. Please confirm that this does not alter your adherence to all PLOS ONE policies on sharing data and materials, by including the following statement: "This does not alter our adherence to  PLOS ONE policies on sharing data and materials.” (as detailed online in our guide for authors http://journals.plos.org/plosone/s/competing-interests).  If there are restrictions on sharing of data and/or materials, please state these.

Please note that we cannot proceed with consideration of your article until this information has been declared.

Reviewers' comments:

Reviewer's Responses to Questions

**Comments to the Author**

1. Is the manuscript technically sound, and do the data support the conclusions?

Reviewer #1: Partly

Reviewer #2: Partly

2. Has the statistical analysis been performed appropriately and rigorously? 

Reviewer #1: No

Reviewer #2: I Don't Know

3. Have the authors made all data underlying the findings in their manuscript fully available?

Reviewer #1: Yes

Reviewer #2: Yes

4. Is the manuscript presented in an intelligible fashion and written in standard English?

Reviewer #1: Yes

Reviewer #2: No

5. Review Comments to the Author

Reviewer #1: The primary analysis used was the proportional hazard regression models which were fitted to quantify effects of predictors on time to recovery. The Kaplan Meier plots were used to visualize results concerning age and admission categories. Restricted mean time to recovery was estimated for significant categorical predictors. Logistic regression was used to evaluate predictors of non-response to treatment. Both unadjusted and adjusted models for sex and age at admission were fitted. The analysis appears reasonable.

The data is from the MANGO study . The study design and some protocol information is noted on page 4. The authors note on page 7 that excluding the 157 referrals, 644 children contributed to the analysis of time to recovery and all 801 to the analysis of non-response to treatment. The study design lacks statistical justification in the requires sample size from a power perspective. Given the large number of variables considered in the time to event and non response analyses and the large number of p-values presented, some rationale for the sample size is appropriate, despite the descriptive nature of the study. The sample size certainly appears adequate. However, the authors should comment on the statistical aspects of its adequacy.

Reviewer #2: This study was an offshoot study from the Mango trial looking at the effect of a nutrient supplement on recovery from severe acute malnutrition

Firstly, the paper contains a number of grammatical errors that needs to be addressed.

Abstract/Introduction

More details need to be added on what SAM is, what is known about the causes of SAM etc.

What is not clear is if SAM can be caused by socioeconomic factors such as lack of food availability. This needs to be addressed as it ties into some of the speculations made in the discussion section in relation to categorizing the participants that present with illness vs those that don’t and their recovery.

4 million children where? Globally or locally?

Methods/results

It is unclear whether the data analysed are regarding standard RUTF or reduced RUTF

Also what is RUTF? A Nutrition label table should be added.

Equations of WHZ and MUAC should be included

Line 81: what does subsist mean?

Clearer detail on the study protocol should be included: Number of visits, when treatment was administered, dose, information collected, etc—in a figure.

Line 93: a reference needs to be included for the “national protocol”

Details of manufacturer for the kits/equipment used needs to be included in the “data collection” section

Line 108-109 how was data monitoring and cleaning performed?

Not clear on how the predictor values were calculated in terms of hazards ratio, REF and odds ratio. Needs to be clarified.

What is open defecation?

6. PLOS authors have the option to publish the peer review history of their article (what does this mean?). If published, this will include your full peer review and any attached files.

Reviewer #1: No

Reviewer #2: No

---

## [Author Response · Author response to Decision Letter 0]

30 May 2021

Dear Editor and Reviewers,

Many thanks for the careful review of our manuscript on the “Predictors of time to recovery and non-response during outpatient treatment of severe acute malnutrition”. We truly appreciated the thoughtful feedback provided and have now worked to improve our manuscript addressing the specific points raised. Please find below an explanation of the improvements made to the manuscript. Our responses are highlighted in bold italics.

Editors comments:

Response: Thank you for the orientation. We have now revised the formatting to correspond to the Plos One requirements.

'STK was previously employed by Nutriset, a producer of RUTF. HF has received research grants from ARLA Food for Health Centre, and also has research collaboration with Nutriset. Other authors declare no financial relationships

with any organisations that might have an interest in the submitted work in the previous five years, and declare no other relationships or activities that could appear to have influenced the submitted work.'

a. Please confirm that this does not alter your adherence to all PLOS ONE policies on sharing data and materials, by including the following statement: "This does not alter our adherence to PLOS ONE policies on sharing data and materials.” (as detailed online in our guide for authors http://journals.plos.org/plosone/s/competing-interests). If there are restrictions on sharing of data and/or materials, please state these.

Please note that we cannot proceed with consideration of your article until this information has been declared.

Response: Thank you for this guidance. We have now revised the conflict of interest statement to include the sentence “The declared competing interests do not alter our adherence to PLOS ONE policies on sharing data and materials”.

Response: Thank you, this has been done.

Response: Thank you for this clarification. We have thus pursued to suppressing this part of the discussion: More children with both admission criteria reached MUAC ≥125 mm before reaching a WHZ ≥-2 (data not shown). It would thus seem that reaching a WHZ ≥-2 is more challenging for recovering young children and maybe unreachable for some as indicated by the high proportion of non-response to treatment. 

Response: Thank you, this has now been done.

Reviewers' comments:

5. Review Comments to the Author

Reviewer #1: The primary analysis used was the proportional hazard regression models which were fitted to quantify effects of predictors on time to recovery. The Kaplan Meier plots were used to visualize results concerning age and admission categories. Restricted mean time to recovery was estimated for significant categorical predictors. Logistic regression was used to evaluate predictors of non-response to treatment. Both unadjusted and adjusted models for sex and age at admission were fitted. The analysis appears reasonable.

The data is from the MANGO study . The study design and some protocol information is noted on page 4. The authors note on page 7 that excluding the 157 referrals, 644 children contributed to the analysis of time to recovery and all 801 to the analysis of non-response to treatment. The study design lacks statistical justification in the requires sample size from a power perspective. Given the large number of variables considered in the time to event and non response analyses and the large number of p-values presented, some rationale for the sample size is appropriate, despite the descriptive nature of the study. The sample size certainly appears adequate. However, the authors should comment on the statistical aspects of its adequacy.

Response: Thank you for raising this question. We studied a total of 25 predictors for a total of 433 events resulting in ratio of 17 events per predictor (EPP). Often 10 EPP is considered sufficiently robust for predictor analysis. When calculating the power it seems we have a 64% power to detect a 20% increase or reduction in the hazard (HR<0.8 or HR>1.25). While this post hoc power calculation can be done it does not seem like good statistical practice (Hoenig & Heisey, 2001 ). So instead in the manuscript we have now given reference to the sample size of similar studies that, in most cases, seem to include a sample of around 400 children treated. This is how it states now in the manuscript: “The study recruited a total of 801 children, which was an adequate sample size for analyzing predictors as judged by previous studies looking at time to recovery with samples sizes starting from 200 and most around 400 children treated [16–27].” (LINE 83-90 in final version without trackchanges)

Reviewer #2: This study was an offshoot study from the Mango trial looking at the effect of a nutrient supplement on recovery from severe acute malnutrition

Firstly, the paper contains a number of grammatical errors that needs to be addressed.

Response: Thank you for this note. We have now revised the manuscripts English thoroughly.

Abstract/Introduction

More details need to be added on what SAM is, what is known about the causes of SAM etc.

What is not clear is if SAM can be caused by socioeconomic factors such as lack of food availability. This needs to be addressed as it ties into some of the speculations made in the discussion section in relation to categorizing the participants that present with illness vs those that don’t and their recovery.

Response: Thank you for this comment. We have now included a paragraph on the causes of SAM in the Introduction section as follows: “Severe acute malnutrition (SAM) is a condition that occurs when the food intake does not meet the nutritional requirements either as a consequence of poor intake or disease [1]. Generally SAM arises in contexts with social, political and economic factors affecting food availability and where infections and inflammation are common [1]. This is also why no single intervention has been shown to reduce the incidence that requires a holistic package of interventions [2].” (LINE 43-47 in final version without trackchanges)

4 million children where? Globally or locally?

Response: Thank you for this comment. We now added the precision of “globally” in the following sentence: “Every year, more than 4 million children are treated for SAM globally [6] with a varying mean treatment time and proportion of non-response to treatment [7–11].” (LINE 57-58 in final version without trackchanges)

Methods/results

It is unclear whether the data analysed are regarding standard RUTF or reduced RUTF

Response: Thanks for the question. The data were analysed for all children treated regardless of their treatment arm. This has now been more clearly stated in the Methods section as follows: “All children included in the MANGO trial regardless of their treatment arm (reduced RUTF or standard RUTF) were included in the analysis.” (LINE 157-158 in final version without trackchanges)

Also what is RUTF? A Nutrition label table should be added.

Response: Thanks for the question. We have now added the following to the introduction section: “RUTF are energy and nutrient dense pastes usually composed of peanut butter, milk powder, oil, sugar and a vitamin and mineral complex designed to fulfil the nutritional needs of children recovering from SAM [4].” (LINE 51-53 in final version without trackchanges)

Equations of WHZ and MUAC should be included

Response: Thanks for the helpful comment. We have now added these precisions to the methods section under Data collection: “Weight was measured using an electronic scale (SECA 876) to the nearest 100 g, height using a wooden measuring board (locally made) to the nearest 1 mm, and MUAC using a non-stretchable colourless measuring tape to the nearest 1 mm. Z-scores were calculated using the WHO standards and STATA command zscore06 [21].” (LINE 115-118 in final version without trackchanges)

Line 81: what does subsist mean?

Response: Thanks for noting. We have changed the word to “depend” (LINE 100 in final version without trackchanges)

Clearer detail on the study protocol should be included: Number of visits, when treatment was administered, dose, information collected, etc—in a figure.

Response: Thanks for pointing out. We have added some clarifications for these points in the Methods section under the Study participants and treatment protocol and Data collection sub-sections. They detail when and what treatment was administered (specifically at admission and then during the weekly follow ups) and the type of data collected at different time points.

Line 93: a reference needs to be included for the “national protocol”

Response: Thank you. This has been done.

Details of manufacturer for the kits/equipment used needs to be included in the “data collection” section

Response: Thank you. These details have been included and the following additions/edits made to the sections: “Weight was measured using an electronic scale (SECA 876) to the nearest 100 g, height using a wooden measuring board (locally made) to the nearest 1 mm, and MUAC using a non-stretchable colourless measuring tape to the nearest 1 mm. Z-scores were calculated using the WHO standards and STATA command zscore06 [21].” (LINE 115-118 in final version without trackchanges)

Line 108-109 how was data monitoring and cleaning performed?

Response: Thank you for the comment. We have now added the following section to the Data collection sub-section of the Methods: “Data monitoring included among other thing, checking duplicate entries and outliers, anthropometric decimal distributions and correct prescription of medication according to diagnosed conditions. Any potential data problem resulted in action. Data cleaning was based on double checks of electronic data against patient registries or therapeutic cards.” (LINE 120-123 in final version without trackchanges)

Not clear on how the predictor values were calculated in terms of hazards ratio, REF and odds ratio. Needs to be clarified.

Response: Thank you for mentioning this shortcoming. We have now clarified this in the Data analysis sub-section of the Methods as follows: “Cox proportional hazard regression models were fitted to quantify effects of predictors on time to recovery by means of hazard ratios, describing the increased or decreased rate of change in likelihood of recovery.” (LINE 159-161 in final version without trackchanges)

What is open defecation?

Response: Thank you for the question. We have added the definition of open defecation in the Methods now as follow: “open defecation (the practice of defecating in nature instead of a toilet facility)” (LINE 150 in final version without trackchanges)

---

## [Decision Letter · Decision Letter 1]

9 Feb 2022

PONE-D-20-35912R1Predictors of time to recovery and non-response during outpatient treatment of severe acute malnutritionPLOS ONE

Dear Dr. Kangas,

Thank you for submitting your manuscript to PLOS ONE. We feel that the manuscript is almost acceptable for publication, but that some minor points need to be addressed first. Therefore, we invite you to submit a revised version of the manuscript that addresses the points raised during the review process.

In addition to the comments by Reviewer 3, could you please take note of the following comments:

Line 33. Suggest changing to  ..<-3, anaemia or no illness at admission…’  as it is now not clear whether the ‘no’ refers only to illness or also to anaemia.

Line 169-173. Can the authors explain the difference between ‘non-response to treatment’, which to me would indicate lack of weight gain, and ‘referred to inpatient care due to stagnant weight or weight loss’ (I leave out here the medical complications, which are different). Would the results have been different if children referred to inpatient care because of lack of weight gain would have been included in the analysis of time to recovery?

In table 4. How can sex and age still be in the model when the model was adjusted for these predictors (last columns)

We look forward to receiving your revised manuscript.

Kind regards,

Frank Wieringa, M.D., Ph.D.

Academic Editor

PLOS ONE

Journal Requirements:

Reviewers' comments:

Reviewer's Responses to Questions

**Comments to the Author**

1. If the authors have adequately addressed your comments raised in a previous round of review and you feel that this manuscript is now acceptable for publication, you may indicate that here to bypass the “Comments to the Author” section, enter your conflict of interest statement in the “Confidential to Editor” section, and submit your "Accept" recommendation.

Reviewer #1: All comments have been addressed

Reviewer #3: (No Response)

2. Is the manuscript technically sound, and do the data support the conclusions?

Reviewer #1: (No Response)

Reviewer #3: Yes

3. Has the statistical analysis been performed appropriately and rigorously? 

Reviewer #1: (No Response)

Reviewer #3: Yes

4. Have the authors made all data underlying the findings in their manuscript fully available?

Reviewer #1: (No Response)

Reviewer #3: Yes

5. Is the manuscript presented in an intelligible fashion and written in standard English?

Reviewer #1: (No Response)

Reviewer #3: Yes

6. Review Comments to the Author

Reviewer #1: (No Response)

Reviewer #3: The study provides relevant information on a subject with limited literature. The publication of the study results could contribute to the current discussion on the need to further decentralise essential health services to reach a broader number of people in need.

Some aspects of this work could have been more clearly presented and explained, as detailed below:

Introduction

The authors should extend the rationale and explain why the topic treated is particularly crucial.

This could be done by better explaining the absolute magnitude of the problem and, on the other hand, by better underling the implications of the study's results on current SAM treatment and prevention programs.

In the background, the authors mentioned the last figure available for coverage without mentioning the global burden of wasted children 6-59months.

The comparison between global burden and treatment coverage shows how coverage is still impressively low. Thus, how it is essential to reach more children affected by SAM. As mentioned in the paper, a shorter treatment length would eventually decrease costs per individual treated. Reduced cost per child treated could subsequently help extend the coverage of the programs.

The authors should also consider explaining that global data reported are only for children 6-59 months and eventually why this target group is so critical.

Methods

Statistical analysis is clearly described.

Discussion

Findings are clearly resumed and discussed.

Minor points:

-Consider eventually modifying the abstract according to the modifications done in the introduction

- line 150, it would be better to specify what acronym ARI is the first time it is used. The authors specified it only a few lines below.

7. PLOS authors have the option to publish the peer review history of their article (what does this mean?). If published, this will include your full peer review and any attached files.

Reviewer #1: No

Reviewer #3: **Yes: **Silvia Barbazza

---

## [Author Response · Author response to Decision Letter 1]

5 Mar 2022

Dear Editor and Reviewers,

Many thanks for the careful review of our manuscript on the “Predictors of time to recovery and non-response during outpatient treatment of severe acute malnutrition”. We truly appreciated the thoughtful feedback provided and have now worked to improve our manuscript addressing the specific points raised. Please find below an explanation of the improvements made to the manuscript. 

Editors comments:

Line 33. Suggest changing to ..<-3, anaemia or no illness at admission…’ as it is now not clear whether the ‘no’ refers only to illness or also to anaemia.

Response: Thanks for the observation. We have now changed the sentence as follows:

LINE 33: “…WHZ <-3, no illness nor anaemia at admission,..”

Line 169-173. Can the authors explain the difference between ‘non-response to treat-ment’, which to me would indicate lack of weight gain, and ‘referred to inpatient care due to stagnant weight or weight loss’ (I leave out here the medical complications, which are different). Would the results have been different if children referred to inpa-tient care because of lack of weight gain would have been included in the analysis of time to recovery?

Response: Thanks for the observation. Indeed we would expect that children ending up as non-responders would present with no weight gain during treatment. However, in our cohort we saw that some children continued to gain weight but with a very slow rate which meant that they were not classified as stagnant weights (where the definition was no more than 100g gained over 4 weeks) but continued their treatment. These children could also be gaining in height meaning that their WHZ was not improving which meant they were not reaching the recovery cut-off of WHZ-2. Also, to clarify the implication on the results of excluding the children referred to inpatient care we firsts added a sentence in the outcome definition section on line 142 and then a sentence on the sensitivity analysis results when including these children (which led to similar results).

Thus, we have now added the following sections:

LINE 101-103: “In case of medical complications, weight loss of over 5% at any point or stagnant weight defined as no more than 100g weight gained over 4 weeks, dur-ing treatment, children were referred to inpatient care, as per the Burkina national CMAM protocol.”

LINE 141-144: “For the study of time to recovery, non-recovered cases were right censored contributing to the analysis of time to recovery until exit from study. Pa-tients referred to inpatient care were excluded from the analysis in order to limit potential bias that could be introduced due to short length of stay in treatment.”

LINE 210: “Including children referred to inpatient care in the analysis of different factors with time to recovery resulted in similar associations.”

In table 4. How can sex and age still be in the model when the model was adjusted for these predictors (last columns)

Response: Thanks for the question. The odds ratios for these variables (sex and age) in the sex and age adjusted model represent the OR adjusted for the other variable. Meaning, when looking at sex as a predictor, the adjusted model adjusts for age to look at the age independent predictive power of sex. And the inverse in the case of age. We have now made this clearer in all the tables (2, 3 and 4) adding a footnote indicating that “when analysing sex as a predictor, only age was included as adjustment and when analysing age as a predictor only sex was included as adjustment”.

6. Review Comments to the Author

Reviewer #3: The study provides relevant information on a subject with limited literature. The publication of the study results could contribute to the current discussion on the need to further decentralise essential health services to reach a broader number of peo-ple in need.

Some aspects of this work could have been more clearly presented and explained, as detailed below:

Introduction

The authors should extend the rationale and explain why the topic treated is particularly crucial. This could be done by better explaining the absolute magnitude of the problem and, on the other hand, by better underling the implications of the study's results on current SAM treatment and prevention programs.

Response: Thanks for the suggestion. We added the following sections highlighting the magnitude of the problem (in the 1st paragraph of the introduction section) and the subsequent coverage of treatment (4th paragraph of the introduction section):

LINE 43 – 49: “Severe acute malnutrition (SAM) is a condition that occurs when the food intake does not meet the nutritional requirements either as a consequence of poor intake or disease [1]. SAM is diagnosed when a child presents with a weight-for-height z-score (WHZ) <-3, a mid-upper-arm circumference (MUAC) <125 mm or nutritional oedema [2]. While the overall prevalence of SAM is unknown, in 2020 it was estimated that 2% of all children below the age of 5 years presented a WHZ <-3 translating to more than 13.6 million children suffering from severe wasting at any time [3]. Children with SAM have a 11.6 increased risk of mortality compared to children with no nutritional deficits living in the same contexts [4].”

LINE 62-66: “In 2020, around 5 million children were treated for SAM globally [10]. With 13.6 million children suffering from severe wasting at any time and applying of incidence correction factor of 3.5 [11] to account for all new cases arising in a year, this translates to 47.6 million episodes of severe wasting in a year. Thus, the current coverage of treatment is around 10%. This is when excluding the burden of MUAC cases which, if added, would translate to an even lower coverage of treatment. Such low coverage warrants for reflection on how to improve it possibly by optimising and better targeting treatment in order to expand it to more cases.”

In the background, the authors mentioned the last figure available for coverage without mentioning the global burden of wasted children 6-59months.

Response: Thanks for the observation. We added the following sections highlighting the magnitude of the problem (in the 1st paragraph of the introduction section) and the subsequent coverage of treatment (4th paragraph of the introduction section):

LINE 43 – 49: “Severe acute malnutrition (SAM) is a condition that occurs when the food intake does not meet the nutritional requirements either as a consequence of poor intake or disease [1]. SAM is diagnosed when a child presents with a weight-for-height z-score (WHZ) <-3, a mid-upper-arm circumference (MUAC) <125 mm or nutritional oedema [2]. While the overall prevalence of SAM is unknown, in 2020 it was estimated that 2% of all children below the age of 5 years presented a WHZ <-3 translating to more than 13.6 million children suffering from severe wasting at any time [3]. Children with SAM have a 11.6 increased risk of mortality compared to children with no nutritional deficits living in the same contexts [4].”

LINE 62-66: “In 2020, around 5 million children were treated for SAM globally [10]. With 13.6 million children suffering from severe wasting at any time and applying of incidence correction factor of 3.5 [11] to account for all new cases arising in a year, this translates to 47.6 million episodes of severe wasting in a year. Thus, the current coverage of treatment is around 10%. This is when excluding the burden of MUAC cases which, if added, would translate to an even lower coverage of treatment. Such low coverage warrants for reflection on how to improve it possibly by optimising and better targeting treatment in order to expand it to more cases.”

The comparison between global burden and treatment coverage shows how coverage is still impressively low. Thus, how it is essential to reach more children affected by SAM. As mentioned in the paper, a shorter treatment length would eventually decrease costs per individual treated. Reduced cost per child treated could subsequently help extend the coverage of the programs.

Response: Indeed! To make this clear in the manuscript, we added the following paragraph in the Introduction section:

LINE 62-66: “In 2020, around 5 million children were treated for SAM globally [10]. With 13.6 million children suffering from severe wasting at any time and applying of incidence correction factor of 3.5 [11] to account for all new cases arising in a year, this translates to 47.6 million episodes of severe wasting in a year. Thus, the current coverage of treatment is around 10%. This is when excluding the burden of MUAC cases which, if added, would translate to an even lower coverage of treatment. Such low coverage warrants for reflection on how to improve it possibly by optimising and better targeting treatment in order to expand it to more cases.”

The authors should also consider explaining that global data reported are only for children 6-59 months and eventually why this target group is so critical.

Response: Thanks for the comment. We added the following sentence to the 1st paragraph of the introduction section to clarify the importance of targeting children with SAM due to their high risk of mortality:

LINE 49: “Children with SAM have a 11.6 increased risk of mortality compared to children with no nutritional deficits living in the same contexts [4].”

Methods

Statistical analysis is clearly described.

Response: Thanks!

Discussion

Findings are clearly resumed and discussed.

Response: Thanks!

Minor points:

-Consider eventually modifying the abstract according to the modifications done in the introduction

Response: Due to very limited space in abstract, we did not add any of the further justifications now included in the introduction section.

- line 150, it would be better to specify what acronym ARI is the first time it is used. The authors specified it only a few lines below.

Response: Thanks for the observation. We defined ARI now at the 1st appearance on LINE 150.

---

## [Editor Report · Decision Letter 2]

12 Apr 2022

Predictors of time to recovery and non-response during outpatient treatment of severe acute malnutrition

PONE-D-20-35912R2

Dear Dr. Kangas,

We’re pleased to inform you that your manuscript has been judged scientifically suitable for publication and will be formally accepted for publication once it meets all outstanding technical requirements.

Kind regards,

Frank Wieringa, M.D., Ph.D.

Academic Editor

PLOS ONE
---

## [Editor Report · Acceptance letter]

20 May 2022

PONE-D-20-35912R2 

Predictors of time to recovery and non-response during outpatient treatment of severe acute malnutrition 

Dear Dr. Kangas:

I'm pleased to inform you that your manuscript has been deemed suitable for publication in PLOS ONE. Congratulations! Your manuscript is now with our production department. 

Kind regards, 

on behalf of

Dr. Frank Wieringa 

Academic Editor

PLOS ONE